# The Associations between Outdoor Playtime, Screen-Viewing Time, and Environmental Factors in Chinese Young Children: The “Eat, Be Active and Sleep Well” Study

**DOI:** 10.3390/ijerph17134867

**Published:** 2020-07-06

**Authors:** Qiang Wang, Jiameng Ma, Akira Maehashi, Hyunshik Kim

**Affiliations:** 1College of Sports Science, Shenyang Normal University, Shenyang 110034, China; 13804999441@163.com; 2Faculty of Physical Education, Sendai University, Miyagi 9891693, Japan; jm-ma@sendai-u.ac.jp; 3Faculty of Human Sciences, Waseda University, Saitama 3591192, Japan; maehashi@waseda.jp; 4Advanced Research Center for Human Sciences, Waseda University, Saitama 3591192, Japan

**Keywords:** outdoor play, screen-viewing time, environment, preschool children, urban/rural

## Abstract

The purpose of this study is to identify regional differences in outdoor activity time and screen-viewing time of preschool children in urban and rural areas and to provide data on the environmental factors to identify modifiable determinants for each region. This cross-sectional study was conducted on 1772 out of 2790 children between the age of 3 to 6 years living in northern China, with their consent. A cross-sectional study was conducted among preschool children living in urban (*n* = 1114) and rural areas (*n* = 658) in northern China. To assess environmental factors, the International Physical Activity Questionnaire was used for neighborhood environments, and the questionnaire included three items each for the physical home environment and socio-cultural environment domains. We observed the associations between outdoor play for urban children and sidewalks in the neighborhood, paths for cycles, aesthetic qualities, and “motor vehicles. In addition, in rural areas, screen-viewing time and environmental factors were found to be positively correlated with traffic, limited place and method of outdoor play, and were negatively correlated with the importance of academics and need for company in outdoor play. This has important implications for the development of effective intervention programs for preschool children in China in the future.

## 1. Introduction

Regular, appropriate outdoor play is an important health-related factor that promotes physical, mental, and social growth in young children [1,2,3]. Outdoor play refers to unorganized, outdoor physical activities. Since it involves higher energy costs compared to resting but lower than exercising [4], the amount of time spent outdoors can be used to evaluate the level of physical activity in young, preschool children and is thus very important [5].

However, significant decreasing trends in children’s outdoor playtime have been reported in the past several decades in the Western world, including countries like the USA [6], Australia [7], and Canada [8]. Previous studies reported that outdoor play, in which children can interact with nature, yields favorable health-related outcomes, such as improved physical abilities and high levels of physical activity in preschool children [9,10]. Nevertheless, changes in surrounding environments and life habits have led to increases in screen-viewing time and decreases in outdoor play.

Long screen-viewing times are highly correlated with obesity [11], cardiovascular diseases [12], and decline in sleep [13] and attention in children [14], even when they satisfy the daily recommended physical activities time. The World Health Organization (WHO), being aware of the risk of sedentary behavior based on screen-viewing, recommends 1 h of screen-viewing for children between the age of 2 and 4 years and no screen-viewing for infants between 0 to 1 year. Many countries including the USA, Canada, and Australia, set screen-viewing time in preschool children as a major public health goal [15,16,17].

Previous findings suggested that screen use patterns continue from childhood into adulthood; therefore, interventions to decrease screen-viewing time in childhood are necessary from a public health perspective [18]. Moreover, in to improve health-related behaviors, such as outdoor play and screen-viewing time in preschool children, experts recommend interventions tailored for specific groups. Further, studies focusing on the place of residence (urban and rural), a determinant that has recently gained interest in socioecological models, is also necessary [5]. While it is reported that there are numerous gaps in the level of physical activity and sedentary activities between children living in urban and rural areas, these differences can be partially supplemented by environmental support [19,20].

Although there is significant research assessing the influence of the place of residence in outdoor play and screen-viewing times in Western countries, similar studies have been lacking in Asian preschool children. The present study aimed to investigate regional differences in outdoor playtime and screen-viewing time in Chinese preschool children between the age of 3 and 6 years and their relationship with environmental factors to identify modifiable determinants in regional differences between urban and rural areas.

## 2. Materials and Methods

### 2.1. Study Design and Participants

The present cross-sectional study utilized data from the “Eat, be active and sleep well” study [21], which was conducted to improve life habits of Asian children. The samples were from the Shenyang and Anshan provinces, in Northern China. According to the 2019 Urban Society and Economy Survey by the Chinese National Statistical Office, the cities were classified considering commercial resource power, traffic circulation development, diversification of lifestyle, future development potential, and participation of citizens. Regarding the difference between class 1 cities and new class 1 cities, traditional megacities such as Beijing, Shanghai, Guangzhou, and Shenzhen were included in class 1 cities, whereas newly developed cities that have reached the population of over 10 million due to rapid urbanization were added to new class 1 cities. The cities were classified into class 1, new class 1, class 2, class 3, and class 4 in terms of their size and commercial attraction. According to these criteria, Shenyang was a new class 1 city (population 11,300,000; size 13,000 km²), and Anshan was a class 3 city (population 1,850,000; size 92,000 km²) [22].

We analyzed the data collected from 1772 participants (53% male, 47% female, questionnaire recovery rate of 63.5%) at five kindergartens in Shenyang (*n* = 1512) and five kindergartens in Anshan (*n* = 1278) with ages between 3 and 6 years whose parents received an explanation of the study and provided written informed consent. The surveys were conducted between September and October 2019.

The study received prior approval from the Sendai University Ethics Committee, Faculty of Sports Science (SU29-22).

### 2.2. Measures

#### 2.2.1. Environmental Factors

To assess neighborhood environments, the environmental module of the International Physical Activity Questionnaire (IPAQ-E) was used [23,24]. The IPAQ-E consists of 17 questions on environmental factors with seven core items, four recommended items, and six optional items. This study used ten core and recommended items. The selected items focused on the access to shops, public transportation, presence of sidewalks and bicycle lanes, access to exercise facilities, crime safety, traffic safety, social environment, aesthetics, and number of motor vehicles in the household. These questions referred to the neighborhood environment, defined as the area where a person could walk within 10 to 15 min from his/her residence. The items were rated on a four-point Likert scale (1 = strongly disagree, 2 = somewhat disagree, 3 = somewhat agree, and 4 = strongly agree). The Chinese version of IPAQ-E has shown good test–retest reliability [25].

The questionnaire included three items in the physical home environment domain. Two items, “there is a TV or a computer in the child’s room” and “we limit the length of time spent watching TV, smartphones, tablets, and computers,” were answered by yes/no. The item “how many media devices (TV, smartphones, tablets, and computers) are available at home?” was an open-ended question.

The socio-cultural environment domain included three items answered on a four-point Likert scale (1 = strongly disagree, 2 = somewhat disagree, 3 = somewhat agree, and 4 = strongly agree): “we limit the place or method of outdoor play as we worry about potential injuries,” “academic abilities are more important than physical development through outdoor play,” and “other people, such as siblings, friends, and coaches, should be present for my child’s outdoor play.”

#### 2.2.2. Outdoor Playtime

For outdoor playtime, the parents were asked to provide the average daily time their children spent outside on weekdays and weekends in the past month. Answer options for outdoor playtime were 0, 1–15, 16–30, 31–60, ≥61 min. The parents also provided the daily average outdoor playtime on weekends and weekdays in written answers. Referring to an outdoor playtime checklist by Burdettes et al., children’s outdoor physical activity was assessed by asking parents how much time before noon, from noon until 6:00 p.m., and after 6:00 p.m. the child spent playing in the yard or street around the house and in the park/playground/outdoor recreation area. To accurately evaluate the daily average outdoor playtime on weekdays, outdoor playtime at kindergartens was also optionally surveyed by contacting kindergarten homeroom teachers. This parental recall measure has been shown to correlate with physical activity levels in preschoolers as measured by an accelerometer [26].

#### 2.2.3. Screen-viewing Time

Screen-viewing time was assessed by asking parents how much time their children spent watching TV and using smartphones, tablets, and computers in the past week using the following questions: (1) how many media devices, including TVs, smartphones, tablets, and computers, are available at home?; (2) how much time on average does your child spend on a day watching TV or videos?; and (3) for how long on average in a day does your child use electronic devices, such as smartphones, tablets, and computers?

Subsequently, parents were asked to indicate the average number of days per week and weekends that their child spent on screen-viewing time base on five options: 0, 1–29, 30–59, 60–119, 120–179, or ≥180 min. To calculate the average time spent on the screen-viewing activities per week or weekend, the number of days per week or weekend the child spent time on activities was multiplied by the mid-category values of the duration of the activity per day. Weekday and weekend values were combined and divided by the total number of days.

#### 2.2.4. Demographics

A questionnaire was used to survey the participants’ demographical variables. Information on the children’s sex, age, ethnicity (Han Chinese and others), and area of residence was obtained from their parents.

### 2.3. Statistical Analysis

Data from 1772 young Chinese children (urban: 1114; rural: 658) who provided complete information on the study variables were analyzed using three models. In the first model, non-parametric tests were performed to examine differences between each of the outdoor playtime, screen-viewing time, and the sex, area, and age. In the second model, differences in environmental variables, such as neighborhood environments, physical home environments, and socio-cultural home environments, were assessed through independent samples t-tests. In the third model, a univariate linear regression analysis on each environmental variable was conducted through forced entry to investigate the relationship between screen-viewing and outdoor playtime and environmental variables in urban and rural areas. Moreover, multivariate models adjusted for sex and age were used, and all variables included in each of the multivariate models were assessed for multi-collinearity, which is prevalent among neighborhood environmental characteristics [27]. All statistical analyses were conducted using IBM SPSS 23.0 (IBM, Armonk, NY, USA), and the level of significance was set to *p* < 0.05.

## 3. Results

### 3.1. Study Population

Table 1 shows the children’s mean outdoor playtime and screen-viewing time. Male children had higher mean outdoor playtime and screen-viewing time than females. Outdoor playtime was higher in children living in urban areas whereas screen-viewing time tended to be higher in those living in rural areas. Moreover, outdoor playtime and screen-viewing time increased with age.

### 3.2. Environment Variables in Urban and Rural Participants

Table 2 shows the three environmental domains and their differences by region (mean and standard deviation). The scores for most variables were higher in urban areas. In the neighborhood environment domain, the scores for “closer walking distance to local shops,” “access to the transit stop,” “paths for cycles,” “crime rates during the night,” “aesthetic qualities,” and “motor vehicles” were significantly higher in urban areas. In the physical home environment domain, the score for the “limiting the time spent accessing media devices” item was higher in rural areas, whereas those living in urban areas owned more media devices. In the socio-cultural home environment domain, the scores for all variables were significantly higher in urban areas.

### 3.3. Environmental Variables Associated with Outdoor Playtime

Multivariate regression analyses were conducted after adjusting for sex and age. In the neighborhood environment domain “sidewalks in neighborhood,” “paths for cycles,” “aesthetic qualities,” and “motor vehicles” had positive associations with outdoor playtime in urban areas. Moreover, in the socio-cultural home environment domain “limiting the place or method of outdoor play” and “importance of academic abilities” were negatively associated with outdoor playtime in urban areas, whereas the “need to have others for outdoor play” had a positive association with outdoor playtime in urban areas. For rural areas, we did not observe any significant correlations (Table 3).

### 3.4. Environmental Variables Associated with Screen-viewing Time

In rural areas, a positive correlation was found between screen-viewing time and “traffic” and “limiting the place or method of outdoor play,” while screen-viewing time was negatively correlated with “importance of academic abilities” and “need to have others for outdoor play.” In urban areas, screen-viewing time was positively correlated with “the number of media devices at home” and “limiting the place or method of outdoor play,” and negatively correlated with “importance of academic abilities” and “need to have others for outdoor play” (Table 4).

## 4. Discussion

The present study is one of the few Asian studies analyzing environmental factors and their relationship with outdoor play and screen-viewing times in preschool children in urban and rural areas. We found that outdoor playtime was significantly associated with neighborhood environment and socio-cultural home environment, even after adjusting for sex and age.

Our findings revealed that outdoor playtime in Chinese children living in cities is influenced by factors such as the existence of “sidewalks in the neighborhood,” “paths for cycles,” and “aesthetic qualities” of the neighborhood environment. This result supports previous study findings on the influence of neighborhood environmental factors in western countries [28,29]. Considering the relationship between outdoor play and health in preschool children, this is an important finding for the development of effective intervention programs. In addition, according to a large-scale study on the relation between outdoor play and physical environment, physical neighborhood characteristics (e.g., green neighborhood type, presence of water, diversity of routes) are related to outdoor activities. The study also suggests that children’s age and sex must be considered to increase their outdoor activity time [30,31]. According to Play Streets, an interventional study aiming to improve environmental factors in the USA and multiple European countries, American children who actively participated in the study activities had three-fold increases in the amount of physical activity, and similar positive effects were reported in Belgium as well [32,33]. In light of these findings, improving neighborhood environmental factors is an important strategy to increase outdoor play in Chinese preschool children living in urban areas.

An interesting finding of this study is that having transportation means (e.g., car and motorcycle) was associated with children’s outdoor playtime. Particularly, for children to enjoy outdoor play safely in the cities, they need to go to large parks or exercise facilities by car. In other words, parental awareness and social support are prerequisites for outdoor activities, especially in urban areas [34]. This is because parents who influence their children’s behavior usually spend a great deal of time with them. Moreover, their ability and motivation to bring children outside influences the children’s opportunities for outdoor play [35].

In addition to the neighborhood environment, factors in the socio-cultural home environment, such as “limiting the place or method of outdoor play,” “importance of academic abilities,” and “need to have others for outdoor play,” were positively associated with outdoor playtime in children living in the cities. As for “sidewalks in the neighborhood,” “paths for cycles,” and “limiting the place or method of outdoor play”, they are related to the safety of the surrounding environment. According to a Canadian study, 82% of parents had concerns about their children’s safety during outdoor play [36]. An Australian longitudinal study conducted between 2007 and 2013 also found that parents concerned about their children playing outdoors increased notably from 26% to 42% [37]. Furthermore, according to studies done by objective measurements, since concerns regarding crimes against children or traffic accidents are large obstacles to outdoor play of children, forming a safe environment in the urban areas is an important method for increasing outdoor activities of Chinese children [38].

Accordingly, when planning urban environments, the establishment of parks or exercise facilities where outdoor activities can take place safely should be considered. Since society now emphasizes mastering academic abilities even in childhood, many more parents value and prioritize academic abilities over physical activities. However, significant studies have reported that physical activity during childhood influences brain development [39]. Therefore, parents should actively participate in outdoor activities with their children, and simple exercises, such as parent-child stretching, should be recommended. Parental education about the benefits of physical activity during childhood might prove useful as well.

In this study, we found an association between screen-viewing time and the “traffic” factor in the neighborhood environment in urban areas; no association was found with other environmental factors. This is consistent with previous research on adolescents in western countries [40,41]. However, in contrast with previous studies, we also found an association between screen-viewing time and traffic in rural areas. This might be due to excessive noise, fast traffic speed, and lack of school zones, which may interfere with preschool children’s outdoor play and, instead, encourage them to spend more time watching TV or playing at home. According to a previous meta-analysis finding, safety measures for urban traffic, such as speed limits and one-way traffic, can decrease the number of injuries by 15% on average, as well as deaths associated with collisions [42,43]. These policies can reassure parents about their children’s safety, which may be an important factor to reduce screen-viewing time and increase outdoor activities in children.

In the physical home environment, there was a significant association between the number of media devices at home and screen-viewing time in children living in urban environments. A systematic review revealed that significant correlation was found in only one out of five studies reported thus far [44]. Moreover, previous studies that separately analyzed urban and rural areas reported that screen-viewing time was higher in urban areas [45,46], whereas others reported no significant difference [47,48]. Since previous studies conducted thus far were done in western countries, it is difficult to apply their findings to Asian populations. In this study, the number of media devices, including TV, smartphones, tablets, and computers, differed in urban and rural areas. In addition, the availability of the internet and the number of media devices tends to be higher in urban areas (74.6%) than in rural areas (26.7%) in China, which would result in higher screen-viewing time in urban areas [49]. Therefore, in urban families, in order to prevent excessive screen use in children, decreasing media access should be a priority when designing interventions.

In the socio-cultural home environment, screen-viewing time was associated with “limiting the place or method of outdoor play,” “importance of academic abilities,” and “need to have others for outdoor play” in both rural and urban areas. In the relationship between this domain and screen-viewing time, the role of parents is particularly important. According to a systematic review, parental participation, rather than perceived neighborhood factors, was found to be an important determinant of successful interventions for preschool children, and statistically significant changes and consistent association with sedentary activities in children were also found [50]. This is because the activities in which children participate depend greatly on their parents: if parents limit their children’s outdoor playtime or limit their activities to home or other places where the children can be easily supervised, children may increasingly habituate to watching TV or playing video games [51,52]. Outdoor playtime, especially with parents, provides children with necessary experiences/ emotions for their development: freedom, fun, creativity, and confidence, thus decreasing screen viewing time.

Several limitations of our study should be considered when interpreting our findings. First, due to the cross-sectional study design, causality could not be confirmed in the association between outdoor playtime and screen-viewing time and environmental factors in urban and rural areas. Longitudinal or interventional studies should be conducted to reveal causal relationships between these variables. Second, since the present study was conducted on preschool children in northeast China, the results may not be generalizable to all preschool children in China. Further studies with different samples should be conducted. Third, as in other studies conducted with preschool children, the questionnaire surveys were completed by parents or family members who observed the children’s behavior. Fourth, this study has limitations due to the use of a questionnaire, a subjective measurement method, rather than using an accelerometer, an objective measurement method, to calculate children’s outdoor activity time.

## 5. Conclusions

The present study re-emphasized the evidence on various environmental factors associated with outdoor play and screen-viewing times in preschool children. Our findings suggest that urban environment planning should consider establishing parks and exercise facilities where children can play safely, as this would reassure parents and increase outdoor playtime. Moreover, improving the neighborhood environment (e.g., creating cycle paths, sidewalks, green areas) will increase outdoor playtime, especially in urban settings. Furthermore, traffic control (e.g., monitoring speed limits) is necessary in urban and rural settings to increase children’s outdoor playtime. Lastly, decreasing children’s access to media, particularly in urban households should be a priority in reducing screen-viewing time. Our study focused on Asian preschool children, a population barely studied in previous research; therefore, the findings have significant practical implications for the development of effective intervention programs, and they contribute to the existent literature on the topic.

## Figures and Tables

**Table 1 ijerph-17-04867-t001:** Median of outdoor playtime and screen viewing time minutes per week for sex, areas and age (*n* = 1772).

Variables	*N* (%)	Outdoor Playtime (min/day)	Screen View Time (min/day)
Sex			
Boy	941 (53)	93.24 ^a^	111.60 ^a^
Girl	831 (47)	87.53	98.93
Area			
Urban	1114 (63)	94.47 ^b^	101.68
Rural	658 (37)	83.51	113.24 ^c^
Age			
3	413 (23)	85.54	101.10
4	610 (34)	88.80	105.11
5	616 (35)	93.01	106.62
6	133 (8)	100.78 ^d^	121.91 ^d^

a: Boy are higher in Outdoor playtime and screen view time more than the girl (*p* < 0.001) b: Urban is more active in Outdoor playtime more than rural (*p* < 0.001) c: Rural is higher in screen view time more than urban (*p* < 0.01) d: The age group of 6 years in Outdoor playtime and screen view time more than the age group of 3–5 (*p* < 0.001).

**Table 2 ijerph-17-04867-t002:** Mean (SD) of environmental correlates for urban (*n* = 1114) and rural (*n* = 658) participants.

Environmental Variables	Urban Area Mean(SD)	Rural Area Mean (SD)	t
Neighborhood environments ^a^			
Many shops, stores, markets, or other places to buy things I need are within easy walking distance of my home.	3.53 (0.8)	3.3 (1.2)	6.29 *
It is within a 10- to 15-min walk to a transit stop from my home.	3.51 (0.6)	3.4 (0.7)	10.12 ***
There are sidewalks on most of the streets in my neighbourhood.	3.47 (0.7)	3.48 (0.4)	0.39
There are facilities to bicycle in or near my neighborhood shared use paths for cycles and pedestrians.	3.26 (0.8)	3.06 (0.9)	7.04 **
My neighborhood has several free or low-cost recreation facilities.	3.17 (0.7)	2.95 (0.8)	1.47
The crime rate in my neighbourhood makes it unsafe to go on walks at night.	3.21 (0.6)	3.09 (0.1)	6.31 *
There is so much traffic on the streets that it makes it difficult or unpleasant to walk in my neighborhood.	2.96 (0.7)	2.82 (0.7)	1.32
I see many people being physically active in my neighborhood.	3.23 (0.7)	3.07 (0.8)	0.11
There are many interesting things to look at while walking in my neighbourhood	2.97 (0.8)	2.72 (0.8)	6.88 **
How many motor vehicles in working order are there at your household? ^c^	1.53 (0.8)	1.26 (0.7)	4.33 *
Physical home environments ^b^			
There is a TV or a computer in the child’s room.	1.58 (1.0)	1.50 (0.5)	0.10
We limit the length of time spent watching TV, smartphones, tablets, and computers.	1.22 (0.4)	1.27 (0.4)	22.16 ***
How many media devices (TV, smartphones, tablets, and computers) are available at home? ^c^	6.02 (2.3)	5.65 (2.1)	4.33 *
Social-culture home environments ^a^			
We limit the place or method of outdoor play as we worry about potential injuries.	1.70 (0.9)	1.52 (0.8)	12.29 ***
Academic abilities are more important than physical development through outdoor play.	3.61 (1.1)	3.56 (1.1)	2.89 *
Other people, such as siblings, friends, and coaches, should be present for my child’s outdoor play.	2.29 (1.0)	2.19 (0.9)	6.57 **

a: Response option: strongly disagree (1), somewhat disagree (2), somewhat agree (3), strongly agree (4). b: Response option: yes (1), no (2). * *p* < 0.05; ** *p* < 0.01; *** *p* < 0.001. SD: standard deviation. Group differences for continuous variables were assessed using t-tests.

**Table 3 ijerph-17-04867-t003:** Results from linear regression analyses of environmental correlates and outdoor playtime for urban and rural participants.

Environmental Variables	Urban Area (*n* = 1114)		Rural Area (*n* = 658)	
	Unadjusted Beta ^a^	Adjusted Beta ^b^	Unadjusted Beta ^a^	Adjusted Beta ^b^
Neighborhood environments ^c^				
Many shops, stores, markets, or other places to buy things I need are within easy walking distance of my home.	0.07 *	0.05	−0.03	
It is within a 10- to 15-min walk to a transit stop from my home.	0.04		0.02	
There are sidewalks on most of the streets in my neighbourhood.	0.09 *	0.07 *	0.06	
There are facilities to bicycle in or near my neighborhood shared use paths for cycles and pedestrians.	0.13 ***	0.12 ***	0.01	
My neighborhood has several free or low-cost recreation facilities.	0.04		0.05	
The crime rate in my neighbourhood makes it unsafe to go on walks at night.	0.06 *	0.05	0.07	
There is so much traffic on the streets that it makes it difficult or unpleasant to walk in my neighborhood.	0.01		0.07	
I see many people being physically active in my neighborhood.	0.06 *	0.06	0.03	
There are many interesting things to look at while walking in my neighbourhood	0.09 *	0.08 *	0.08 *	0.07
How many motor vehicles in working order are there at your household? ^e^	0.12 ***	0.11 ***	0.04	
Physical home environments ^d^				
There is a TV or a computer in the child’s room.	−0.01		−0.02	
We limit the length of time spent watching TV, smartphones, tablets, and computers.	−0.06		−0.04	
How many media devices (TV, smartphones, tablets, and computers) are available at home? ^c^	−0.08		−0.06	
Social-culture home environments ^c^				
We limit the place or method of outdoor play as we worry about potential injuries.	−0.15 ***	−0.13 ***	0.14	
Academic abilities are more important than physical development through outdoor play.	−0.14 ***	−0.14 ***	0.15	
Other people, such as siblings, friends, and coaches, should be present for my child’s outdoor play.	0.11 **	0.09 *	0.02	

a: Bivariate standardized regression coefficients for each of the environmental variables. b: Adjusted standardized regression coefficients for age (continuous), sex and each of the environmental variables. c: Response option: strongly disagree (1), somewhat disagree (2), somewhat agree (3), strongly agree (4). d: Response option: yes (1), no (2). e: Open-ended response open. * *p* < 0.05; ** *p* < 0.01; *** *p* < 0.001.

**Table 4 ijerph-17-04867-t004:** Results from linear regression analyses of environmental correlates and screen-viewing time for urban and rural participants.

Environmental Variables	Urban Area (*n* = 1114)		Rural Area (*n* = 658)	
	Unadjusted Beta ^a^	Adjusted Beta ^b^	Unadjusted Beta ^a^	Adjusted Beta ^b^
Neighborhood environments ^c^				
Many shops, stores, markets, or other places to buy things I need are within easy walking distance of my home.	−0.01		0.04	
It is within a 10- to 15-min walk to a transit stop from my home.	−0.04		−0.01	
There are sidewalks on most of the streets in my neighbourhood.	−0.06		−0.04	
There are facilities to bicycle in or near my neighborhood shared use paths for cycles and pedestrians.	0.04		0.03	
My neighborhood has several free or low-cost recreation facilities.	0.02		0.01	
The crime rate in my neighbourhood makes it unsafe to go on walks at night.	0.01		0.02	
There is so much traffic on the streets that it makes it difficult or unpleasant to walk in my neighborhood.	0.05		0.11 **	0.10 *
I see many people being physically active in my neighborhood.	0.04		0.02	
There are many interesting things to look at while walking in my neighbourhood	0.02		0.06	
How many motor vehicles in working order are there at your household? ^e^	0.04		0.01	
Physical home environments ^d^				
There is a TV or a computer in the child’s room.	−0.03		−0.04	
We limit the length of time spent watching TV, smartphones, tablets, and computers.	0.07		0.05	
How many media devices (TV, smartphones, tablets, and computers) are available at home?	0.18 ***	0.17 ***	0.08	
Social-culture home environments ^c^				
We limit the place or method of outdoor play as we worry about potential injuries.	0.08 **	0.07 *	0.13 **	0.12 **
Academic abilities are more important than physical development through outdoor play.	−0.11 ***	−0.10 **	−0.12 **	−0.11 **
Other people, such as siblings, friends, and coaches, should be present for my child’s outdoor play.	−0.08 *	−0.07 *	−0.11 **	−0.10 **

a: Bivariate standardized regression coefficients for each of the environmental variables. b: Adjusted standardized regression coefficients for age (continuous), sex and each of the environmental variables. c: Response option: strongly disagree (1), somewhat disagree (2), somewhat agree (3), strongly agree (4). d: Response option: yes (1), no (2). e: Open-ended response open. * *p* < 0.05; ** *p* < 0.01; *** *p* < 0.001.

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
