# Peer review of "The Associations between Outdoor Playtime, Screen-Viewing Time, and Environmental Factors in Chinese Young Children: The “Eat, Be Active and Sleep Well” Study"

_ijerph, 2020, doi:10.3390/ijerph17134867_

Round 1

Reviewer 1 Report

First of all, congratulations for the research that sounds to be integrated in a more broaden study.

The abstract should be improve in order to specify  the data collection and analysis.

The works of Arts and colleagues should be included in the introduction or/and discussion, for example:

Aarts, M., de Vries, S., van Oers, H., & Schuit, A. (2012). Outdoor play among children in relation to neighborhood characteristics: A cross-sectional neighborhood observation study. International Journal of Behavioral Nutrition and Physical Activity, 9, 98. doi:10.1186/1479-5868-9-98

Aarts, M., Wendel-Vos, W., van Oers, H., van de Goor, I., & Schuit, A. (2010). Environmental determinants of outdoor play in children: A large-scale cross-sectional study. American Journal of Preventive Medicine, 39, 212–219. doi:10.1016/j.amepre.2010.05.008

Also some actual references/sources should be added, as for example: Guy Faulkner, Raktim Mitra, Ron Buliung, Caroline Fusco & Michelle Stone (2015). Children's outdoor playtime, physical activity, and parental perceptions of the neighbourhoodenvironment, International Journal of Play, 4:1, 84-97, DOI: 10.1080/21594937.2015.1017303

In methodology section there are few information about the other two surveys. Which was the theoretical support of the surveys? its dimensions, questions, and validation?

The discussion could be improved with some interesting research as I recommend above.

In the conclusions I suggest to highlight the positive association with “the need to have others to outdoor playtime” in urban areas.and the implications of this to effective intervention programs.

Author Response

Reviewer 1:

  1. The abstract should be improve in order to specify the data collection and analysis.

Thank you very much for your comment.

This cross-sectional study was conducted on 1,772 out of 2,790 children (1,114 from urban areas and 658 from rural areas) between the age of 3 to 6 years living in northern China in 2019, with their consent.

The statistics were analyzed through a multivariate linear regression analysis after adjusting for sex and age.

(Please see Abstract, lines 19-21, 26-27 of our revised manuscript).

  1. The works of Arts and colleagues should be included in the introduction or/and discussion.
  2. The discussion could be improved with some interesting research as I recommend above. In the conclusions I suggest to highlight the positive association with “the need to have others to outdoor playtime” in urban areas. and the implications of this to effective intervention programs.

Thank you very much for your comment.

We appreciate your important literature suggestions to support our study. Those papers were examined before initiating our study, while reviewing studies on effects of environmental factors on outdoor activities and screen-viewing time. However, they were excluded due to the age gap between the participants. Nonetheless, we have examined the papers once again and have found that they support our results. Thank you again.

(Please see manuscript, lines 214-218, 239-242 of our revised manuscript).

  1. In methodology section there are few information about the other two surveys. Which was the theoretical support of the surveys? its dimensions, questions, and validation?

Thank you very much for valuable question.

First, regarding the theoretical background of the physical home environment items, we selected three items by extracting papers with positive results in a review paper on the screen-viewing time of preschool children published by our research team. (Kim HS, A.; Ma JM, B.; Maehashi A, C. Factors impacting levels of physical activity and sedentary behavior among young children: a literature review. Int J Appl Sports Sci 2017, 29(1);1-12.) For the background of the socio-cultural environment items, we selected three items in the “Nationwide Survey on Physical Activities and Exercises” published by the Chinese Ministry of Education to promote outdoor activities of young children.

Reviewer 2 Report

Dear Authors,

I read the manuscript with a great interest, because the topic is very relevant and current.

Below you will find some comments to strenghten the manuscript.

Abstract: 

Please, specify the age range for preschool children in China as it might be some differences worldwide.

Conclusions: As the study is conducted in China I strongly suggest to limit the final conclusion to China and not Asian countries. 

Introduction:

Lines 50-52: Please, add the most recent recommendations by WHO, 2019. The World Health Organization (WHO) suggests greater screen-time restrictions: up to 1 hour per day for children from 2 to 4 years old, while in the younger age group (0-1 years old) no screen time is recommended.

Line 69: Please, add the age range for preschool children in China.

Materials and methods:

Lines 78-81: I do not understand why there is class 1 and new class 1? What is the difference? In the case of differences why the same number (1) has been used? I do not quite understand why there were 4 classes and only two cities? Please, try to make it more clear for reader.

What was the procedure to select kindergartens?

Lines 113-114: Do I understand correctly that these were the only options for an answer? That is, the maximum answer was ≥ 60 minutes. Why were the respondents not allowed to set the time themselves? The same question applies for screen-viewing time.

General question: Did the selection method and size of the group allow to obtain a representative group for the preschool population in China?

Results:

Table 1: The description of the results includes statement about longer outdoor time in the urban group. However, there are no statistical tests to support this statement.

Table 2. Please indicate what statistic test was conducted for comparisons of urban and rural area.

Table 3 and 4: The results presented in the table are not clear for the reader (missing values?). Please try to explain more this part.

Discussion:

General remarks: In the discussion, I lacked informations regarding the average time children spent in kindergarten (i.e. how much of the average time per day is spent by child in kindergarten). Information would be interesting regarding whether the way of spending time on "pre-school" and "home" days varied significantly. Could the authors refer to this?

I also wonder if in the case of assessing the impact of environmental factors on screen time and outdoor time, there was a need to analyze how children are spending time while in kindergarten?

Conclusions: 

I have the impression that the conclusion linking urban traffic and speed limits with outdoor and screen-time is not supported by the results of the study.

Author Response

Reviewer 2:

Abstract:

  1. Please, specify the age range for preschool children in China as it might be some differences worldwide.

Thank you very much for your comment.

This cross-sectional study was conducted on 1,772 out of 2,790 children (1,114 from urban areas and 658 from rural areas) between the age of 3 to 6 years living in northern China in 2019, with their consent.

 (Please see Abstract, lines 19-21 of our revised manuscript).

  1. Conclusions: As the study is conducted in China I strongly suggest to limit the final conclusion to China and not Asian countries.

Thank you very much for your comment.

Asian countries will be changed to China.

(Please see Abstract, lines 32 of our revised manuscript).

Introduction:

  1. Lines 50-52: Please, add the most recent recommendations by WHO, 2019. The World Health Organization (WHO) suggests greater screen-time restrictions: up to 1 hour per day for children from 2 to 4 years old, while in the younger age group (0-1 years old) no screen time is recommended.

→Thank you very much for your comment.

The World Health Organization (WHO), being aware of the risk of sedentary behavior based on screen-viewing, recommends 1 hour of screen-viewing for children between the age of 2 to 4 years and no screen-viewing for infants between the age of 0 to 1 year. Many countries including the USA, Canada, Australia, set screen-viewing time in preschool children as a major public health goal.

(Please see Introduction, lines 54-58 of our revised manuscript).

  1. Line 69: Please, add the age range for preschool children in China.

→Thank you very much for your comment.

The present study aimed to investigate regional differences in outdoor playtime and screen-viewing time in Chinese preschool children between the age of 3 to 6 years as subjects and their relationship with environmental factors to identify modifiable determinants in regional differences between urban and rural areas.

(Please see Introduction, lines 70-73 of our revised manuscript).

Materials and methods:

  1. Lines 78-81: I do not understand why there is class 1 and new class 1? What is the difference? In the case of differences why the same number (1) has been used? I do not quite understand why there were 4 classes and only two cities? Please, try to make it more clear for reader.

Thank you very much for valuable question.

Unlike our intention, it seems that the meaning of the content was not delivered thoroughly. According to the reviewer’s advice, we will revise the content to make it simpler and more appealing for our readers.

According to the 2019 Urban Society and Economy Survey by the Chinese National Statistical Office, cities were classified considering commercial resource power, traffic circulation development, diversification of lifestyle, future development potential, and participation of citizens. Regarding the difference between class 1 cities and new class 1 cities, traditional megacities such as Beijing, Shanghai, Guangzhou, Shenzhen were included in class 1 cities, whereas newly developed cities that have reached the population of over 10 million due to rapid urbanization were added to new class 1 cities.

(Please see Materials and methods, lines 78-84 of our revised manuscript).

  1. What was the procedure to select kindergartens?

Thank you very much for valuable question.

Among the kindergartens that have volunteered to participate in our university’s ongoing project (an International Joint Research on the relevance of factors that affect East Asian Children and their Development), the study was conducted on both urban and rural children from 5 kindergartens that agreed to the purpose of the study.

  1. Lines 113-114: Do I understand correctly that these were the only options for an answer? That is, the maximum answer was ≥ 60 minutes. Why were the respondents not allowed to set the time themselves? The same question applies for screen-viewing time.

→Thank you very much for your comment.

The categories of outdoor playtime (0, 1-15, 16-30, 31-60 or ≥61 minutes) were provided as an option for response to each question. Additionally, regarding the average daily playtime, we allowed weekly and weekend outdoor playtime to be recorded freely.

First, we acknowledge that there was a typo and will correct it to 60 minutes.

Then, since we were not able to objectively measure outdoor activity time and screen-viewing time, multiple choice and free entry format were used to prepare the questionnaire with careful consideration of validity.

(Please see Materials and methods, lines 120 of our revised manuscript).

  1. General question: Did the selection method and size of the group allow to obtain a representative group for the preschool population in China?

Thank you very much for valuable question.

For this study, we investigated kindergartens located in the city of Shenyang and Anshan that were participating in the project organized by our university through a cluster sample. The subjects were administrative units such as kindergartens rather than individuals, and basically, every member of the group was selected as a sample. To supplement the weakness of simple random sampling, approximately 2,000 samples were selected and therefore, we believe a representative sample of kindergartens in China have been obtained.

  1. Table 1: The description of the results includes statement about longer outdoor time in the urban group. However, there are no statistical tests to support this statement.

→Thank you very much for your comment.

Following the reviewer’s advice, we presented a new statistical result. Please confirm.

(Please see Results, lines 164-166 of our revised manuscript).

  1. Table 2. Please indicate what statistic test was conducted for comparisons of urban and rural area.

→Thank you very much for your comment.

Following the reviewer’s advice, we presented the statistical method in our text and tables.  Please confirm.

(Please see Results, lines 148-157 of our revised manuscript).

  1. Table 3 and 4: The results presented in the table are not clear for the reader (missing values?). Please try to explain more this part.

Thank you very much for valuable question.

As described in the text, a univariate linear regression analysis and a univariate linear regression analysis after adjusting sex and age have been conducted. To make the table 3 and 4 more compact and accessible to readers and considering the distinctiveness of the study, only the factors that show significant relevance after the univariate analysis were put into a univariate linear regression analysis and recorded in the table.

Discussion:

  1. General remarks: In the discussion, I lacked informations regarding the average time children spent in kindergarten (i.e. how much of the average time per day is spent by child in kindergarten). Information would be interesting regarding whether the way of spending time on "pre-school" and "home" days varied significantly. Could the authors refer to this?

I also wonder if in the case of assessing the impact of environmental factors on screen time and outdoor time, there was a need to analyze how children are spending time while in kindergarten?

→Thank you very much for your comment.

Thank you for your valuable question.

For our research, we checked each kindergarten’s daily curriculum, and confirmed that every kindergarten is running a daily 1-hour outdoor exercise. Each kindergarten had its own distinctive program with subjects such as art, music, Chinese language, play, etc.

In addition, even though daily habit is reported as an important factor, the outdoor activity time questionnaire used for this study cannot make a distinction between weekdays and weekends. Therefore, having to recompense a week’s outdoor activity time was a limitation of the study,

In this study, we did not analyze the effects the time spent in kindergarten and at home has on children’s’ outdoor activity time and screen-viewing time as suggested by the reviewer. However, the study is ongoing and we will submit the result once again to the reviewer.

Once again, thank you for your helpful comment.

Conclusions:

  1. I have the impression that the conclusion linking urban traffic and speed limits with outdoor and screen-time is not supported by the results of the study.

→Thank you very much for your comment.

We have added other references to supplement the insufficient content. Please check.

(Please see manuscript, lines 214-218, 239-242 of our revised manuscript).

Reviewer 3 Report

The paper describes a cross-sectional study aimed to explore the association between specific environmental factors and outdoor activity among Chinese preschool children.

This is a very interesting and relevant piece of work for public health. This paper is timely and includes a large sample (N > 1,700 participants). The study is methodically sound, although the text would benefit from an English language editing. The results are properly discussed and overall it is suitable for publication in IJERPH. Additionally, the authors propose interesting suggestions to promote physical activity in community. However, there are some minor issues that should be revised before I recommend acceptance of this paper.

Lines 14-31. Please delete the introductory terms (i.e., purpose, methods, results and conclusions). Please leave a space between the symbols (i.e., n = 1,114 instead of n=1114).

Please leave a space before references throughout the manuscript and use a hyphen when referring to two or more successive numbers (i.e., line 36 = “…in young children [1-3]” instead of “…in young children[1,2,3]”

Lines 44-46. Please add a reference for this sentence.

Given the topic of the study, it would be appropriate to include the WHO recommendations on physical activity among preschool children.

Lines 76-78. The selection criteria is unclear, please add more information regarding the mentioned coefficients (i.e., x0.25, x0.20, etc.).

Lines 82-84. How were these ten kindergartens selected? Were they randomly selected?

Lines 87-88. How was the confidentiality of the participants guaranteed? Were the surveys answered anonymously?

Line 116. Please place reference number 27 in square brackets.

Lines 154-157. Authors state that boys, older children and children living in urban areas spent more time playing outdoor (minute/day), but Table 1 does not offer a statistical backup for this statement. Authors could conduct non-parametric tests to explore statistical differences in outdoor playtime and screen-viewing times between boys and girls, children living in urban or rural areas, and children of different ages.

Lines 262-270. Authors should briefly expand on the limitations of the methods of data collection. Questionnaires may pose further limitations apart from being completed by parents; for instance, it is very difficult to accurately measure physical activity with only questionnaires, so absence of an objective method of daily physical activity assessment is another limitation.

Author Response

Reviewer 3:

Abstract:

  1. Lines 14-31. Please delete the introductory terms (i.e., purpose, methods, results and conclusions). Please leave a space between the symbols (i.e., n = 1,114 instead of n=1114).

→Thank you very much for your comment.

We made revisions following the reviewer’s suggestions.

(Please see manuscript, lines 20, 22, 89, 90, 167, 179, table 3,4 of our revised manuscript).

  1. Please leave a space before references throughout the manuscript and use a hyphen when referring to two or more successive numbers (i.e., line 36 = “…in young children [1-3]” instead of “…in young children[1,2,3]”

→Thank you very much for your comment.

We made revisions following the reviewer’s suggestions.

(Please see manuscript, lines 41, 58 of our revised manuscript).

  1. Given the topic of the study, it would be appropriate to include the WHO recommendations on physical activity among preschool children.

→Thank you very much for your comment.

The World Health Organization (WHO), being aware of the risk of sitting behavior based on screen-viewing, recommends 1 hour of screen-viewing for children between the age of 2 to 4 years and no screen viewing for infants between the age of 0 to 1 year.

(Please see manuscript, lines 54-57 of our revised manuscript).

  1. Lines 76-78. The selection criteria is unclear, please add more information regarding the mentioned coefficients (i.e., x0.25, x0.20, etc.).

→Thank you very much for your comment.

Unlike our intention, it seems that the meaning of the content was not delivered thoroughly. According to the reviewer’s advice, we will revise the content to make it simpler and more appealing for our readers.

According to the 2019 Urban Society and Economy Survey by the Chinese National Statistical Office, cities were classified considering commercial resource power, traffic circulation development, diversification of lifestyle, future development potential, and participation of citizens. Regarding the difference between class 1 cities and new class 1 cities, traditional megacities such as Beijing, Shanghai, Guangzhou, Shenzhen were included in class 1 cities, whereas newly developed cities that have reached the population of over 10 million due to rapid urbanization were added to new class 1 cities.

(Please see Materials and methods, lines 78-84 of our revised manuscript).

  1. Lines 82-84. How were these ten kindergartens selected? Were they randomly selected?

Thank you very much for valuable question.

Among the kindergartens that have volunteered to participate in our university’s ongoing project (an International Joint Research on the relevance of factors that affect East Asian Children and their Development), the study was conducted on both urban and rural children from 5 kindergartens that agreed to the purpose of the study.

  1. Lines 87-88. How was the confidentiality of the participants guaranteed? Were the surveys answered anonymously?

Thank you very much for valuable question.

This study was conducted in accordance with the affiliated university’s code of ethics and the questionnaire was filled out anonymously. The data is strictly managed by numbering the subjects.

  1. Line 116. Please place reference number 27 in square brackets.

→Thank you very much for your comment.

We have made modifications following the reviewer’s suggestions.

(Please see manuscript, lines 122 of our revised manuscript).

  1. Lines 154-157. Authors state that boys, older children and children living in urban areas spent more time playing outdoor (minute/day), but Table 1 does not offer a statistical backup for this statement. Authors could conduct non-parametric tests to explore statistical differences in outdoor playtime and screen-viewing times between boys and girls, children living in urban or rural areas, and children of different ages.

→Thank you very much for your comment.

Following the reviewer’s advice, we presented a new statistical result. Please confirm.

(Please see Results, lines 164-166 of our revised manuscript).

  1. Lines 262-270. Authors should briefly expand on the limitations of the methods of data collection. Questionnaires may pose further limitations apart from being completed by parents; for instance, it is very difficult to accurately measure physical activity with only questionnaires, so absence of an objective method of daily physical activity assessment is another limitation.

→Thank you very much for your comment.

As the reviewer suggested, parents answering on behalf of their children and measuring physical activities by subjective measurements will be recorded as limitations of the study.

 (Please see manuscript, lines 300-302 of our revised manuscript).
